# Development of Smartphone Application for Markerless Three-Dimensional Motion Capture Based on Deep Learning Model

**DOI:** 10.3390/s22145282

**Published:** 2022-07-14

**Authors:** Yukihiko Aoyagi, Shigeki Yamada, Shigeo Ueda, Chifumi Iseki, Toshiyuki Kondo, Keisuke Mori, Yoshiyuki Kobayashi, Tadanori Fukami, Minoru Hoshimaru, Masatsune Ishikawa, Yasuyuki Ohta

**Affiliations:** 1Digital Standard Co., Ltd., Osaka 536-0013, Japan; y.aoyagi@digital-standard.com; 2Department of Neurosurgery, Shiga University of Medical Science, Otsu 520-2192, Japan; 3Department of Neurosurgery, Nagoya City University Graduate School of Medical Science, Nagoya 467-8601, Japan; 4Normal Pressure Hydrocephalus Center, Rakuwakai Otowa Hospital, Kyoto 607-8062, Japan; rakuwadr1001@rakuwadr.com; 5Interfaculty Initiative in Information Studies/Institute of Industrial Science, The University of Tokyo, Tokyo 153-8505, Japan; 6Shin-Aikai Spine Center, Katano Hospital, Katano 576-0043, Japan; uedashigeo@yahoo.co.jp (S.U.); hoshimar@katano-hp.or.jp (M.H.); 7Division of Neurology and Clinical Neuroscience, Department of Internal Medicine III, Yamagata University School of Medicine, Yamagata 990-9585, Japan; chi.iseki@gmail.com (C.I.); toshiyuki880620@gmail.com (T.K.); yasuyuki@med.id.yamagata-u.ac.jp (Y.O.); 8School of Medicine, Shiga University of Medical Science, Otsu 520-2192, Japan; ds131814@g.shiga-med.ac.jp; 9Human Augmentation Research Center, National Institute of Advanced Industrial Science and Technology (AIST), Kashiwa II Campus, University of Tokyo, Kashiwa 277-0882, Japan; kobayashi-yoshiyuki@aist.go.jp; 10Department of Informatics and Electronics, Faculty of Engineering, Yamagata University, Yamagata 992-8510, Japan; fukami@yz.yamagata-u.ac.jp; 11Rakuwa Villa Ilios, Rakuwakai Healthcare System, Kyoto 604-8402, Japan

**Keywords:** deep learning, motion tracking, markerless motion capture, quantitative gait assessment, smartphone device

## Abstract

To quantitatively assess pathological gait, we developed a novel smartphone application for full-body human motion tracking in real time from markerless video-based images using a smartphone monocular camera and deep learning. As training data for deep learning, the original three-dimensional (3D) dataset comprising more than 1 million captured images from the 3D motion of 90 humanoid characters and the two-dimensional dataset of COCO 2017 were prepared. The 3D heatmap offset data consisting of 28 × 28 × 28 blocks with three red–green–blue colors at the 24 key points of the entire body motion were learned using the convolutional neural network, modified ResNet34. At each key point, the hottest spot deviating from the center of the cell was learned using the tanh function. Our new iOS application could detect the relative tri-axial coordinates of the 24 whole-body key points centered on the navel in real time without any markers for motion capture. By using the relative coordinates, the 3D angles of the neck, lumbar, bilateral hip, knee, and ankle joints were estimated. Any human motion could be quantitatively and easily assessed using a new smartphone application named Three-Dimensional Pose Tracker for Gait Test (TDPT-GT) without any body markers or multipoint cameras.

## 1. Introduction

Video-recorded gait performance has been recommended for assessing gait in patients with gait disturbance [1,2,3,4]. Gait patterns are characterized as the freezing gait, short-stepped gait, shuffling gait, wide-based gait, instability, gait festination, and spastic gait, among others [1,2,3,4,5]. However, these gait patterns are evaluated subjectively and have no standardized rating system. Previously, pathological gait in several movement disorders and fall risk has been assessed objectively using various quantitative measurement instruments; for instance, a reduced stride length and diminished step height were typical spatiotemporal and kinematic characteristics of gait both in iNPH and Parkinson’s disease [5,6,7,8,9,10,11,12,13]. Chen et al. attempted to identify the gait patterns specific to Parkinson’s disease by capturing the lateral view of the walking silhouettes from monocular video imaging with a decorated corridor setup [6]. Although their method could measure the gait cycle time, stride length, walking velocity, cadence, and step frequency, capturing the lateral view only cannot detect gait asymmetry or kinetic gait parameters, such as foot angle and swing amplitude of the upper limbs. Our previous study on the fluctuations in the three-dimensional (3D) acceleration of the trunk during a 3 m Timed Up and Go Test (TUG) revealed that the volume of the 95% confidence ellipsoid for 3D plots of chronological changes in tri-axial acceleration was important for evaluating the severity of gait disturbance in idiopathic normal pressure hydrocephalus (iNPH) [12]. Based on the founding, we previously released the iPhone application “Hacaro-iTUG” (Digital Standard Co., Ltd., Osaka, Japan) which can be freely downloaded from the Apple store (https://itunes.apple.com/us/app/hacaro-itug/id1367832791?l=ja&ls=1&mt=8, accessed on 29 May 2022) and automatically calculate the iTUG score concurrent with the time and 95%CE volume for the 3D acceleration on iTUG. In another study using an inertial accelerometer built into an iPhone and a free application (SENIOR Quality, Digital Standard Co., Ltd.), patients with iNPH with gait disturbance had a significant reduction in the forward and vertical directions and an increase in the lateral direction in trunk acceleration fluctuations during a 15-foot walking test [13]. However, we could not clearly distinguish the six typical gait patterns of freezing, wide-based, short-stepped, and shuffling gaits, instability, and festination of gait using directional trunk acceleration fluctuations. Therefore, we concluded that directly evaluating the motion of the lower limbs quantitatively, for example, 3D angle and angular speed of the knee joint, using some motion capture systems was necessary to distinguish gait patterns. The gold standard for analyzing human motion is using an optical 3D motion capture system with multipoint cameras and reflective markers, such as the Vicon Motion System. However, this method is expensive and time-consuming for the proper placement of approximately 10 cameras with time synchronization and attaching many reflective markers on the subject’s body surface. Therefore, it is difficult to develop into clinical application, although it can accurately extract kinematic and kinetic gait parameters. As a versatile system for relatively inexpensive markerless human body tracking using a depth sensor, Kinect v2 has been used in the gait analysis [14,15,16]. In addition, several markerless motion capture systems applying 3D optical sensing technology were reported in 2021 [17,18,19]. Baak et al. developed a tool for real-time 3D full-body pose estimation from 2.5-dimensional depth images using some pose databases [17]. Buker et al. reported High-Resolution Depth Net (HRDepthNet) model which retrains the original HRNet for depth images using a convolutional neural network [18]. Gutta et al. reported a markerless foot tracking system, although their system needs six temporally and spatially synchronized cameras to capture depth and color and an 8 × 6 chessboard as a reference coordinate system on the floor plane [19]. However, these systems are complicated to prepare and lack versatility, so they are not widespread in general practice. Recent developments in artificial intelligence (AI) have dramatically improved motion tracking technology, which has a strong potential to popularize 3D motion analysis in clinical practice and research. As video-based two-dimensional (2D) motion tracking software, OpenPose [20] and Kinovea [21] have been widely used in gait analysis. Recently, an OpenPose-based 3D pose estimation from the 2D images was reported [22,23,24]; however, two of three reports proposed 3D coordinate estimation by using OpenPose with multiple synchronized video cameras [23,24], while the other employed a deep learning model using training data from 2D coordinates rather than from 2D images [22]. In the past literature, to our knowledge, no AI model has ever successfully estimated the 3D coordinates of human body motions directly from 2D images.

To further advance “well-being, comfort and health monitoring through with wearable sensors”, smart devices would make it easier for anyone to evaluate their family’s walking at home quantitatively. However, to the best of our knowledge, there have been no reports of 3D motion analysis performed on smartphones alone. Therefore, this study was designed to develop a totally novel smartphone application that can directly estimate 3D coordinates of human motion from 2D images taken by the monocular camera of a smartphone using a novel deep learning model.

The summary of the contribution of the study is to utilize the information and communication technology such as AI and smart devices for the quantitative assessment of movement disorders and fall risks in research and clinical practice. The greatest originality of this study was the usage of more than onr million 2D captured images from the motion of digital humans, 3D humanoid computer graphics (CG) characters, as training data for the deep learning model.

## 2. Materials and Methods

### 2.1. Training Dataset for Deep Learning

We used the Common Objects in Context (COCO) 2017 dataset first as training data of 2D human pose estimation. Additionally, to create the original training dataset of 3D pose tracking, 90 original humanoid CG characters created by ourselves (VRM format) were prepared. All humanoid CG characters had a form (skeleton) annotating the 3D coordinates of the 24 following key points: the center of the body (navel), nose, left and right ears, eyes, shoulders, elbows, wrists, thumbs, middle fingers, hips, knees, ankles, and toes. The cubes surrounding the full bodies of the humanoid characters were determined, all 3D coordinates of the 24 key points were normalized, and 2D images were developed with a size of 448 × 448 pixels of red–green–blue (RGB) color images at a maximum frame rate of 60 frames per second (fps). The videoframe was set to a square with the origin in the depth direction to the center of the character’s body. Therefore, the 3D coordinates of the humanoid models were the relative coordinates based on the center of the body without the ground information in the local coordinate system, not the absolute coordinates in the global coordinate system. In addition, the relative coordinates were transformed so that the length from the center of the body to the center of the head was 1 and the length from the center of the body to the ankle joint was 1, regardless of the height in the relative local coordinate system. By capturing the motion of 90 humanoid characters on the Unity system 2019.4 using a virtual camera, more than one million 2D digital images were created as the original 3D pose tracking dataset for the input training data of deep learning, described in Section 2.3. Figure 1 shows the sample images of the original 3D pose tracking dataset which had three continuous 2D images of one humanoid character dancing and walking. The correct relative 3D coordinates based on the skeleton information of these two 3D datasets were attached as Appendix A (training dataset sample.txt).

### 2.2. Execution Environment

The convolutional neural network was the most established algorithm among various deep learning models for processing data that have a grid pattern. However, plain models of the convolutional neural network have a degradation problem, that is, poor training accuracy as the depth of the network increases. Therefore, we modified and used ResNet34, which is a 34-layer residual convolutional neural network with shortcut connections as a backbone network with PyTorch (version 1.4) on Windows 10 (Figure 2).

As a trained model of 3D coordinates estimation from the 2D digital images on the modified ResNet34, the function expressions with numerous parameters were finally created. To make the function expressions available on iPhone application, we first converted the PyTorch format to the Open Neural Network Exchange (ONNX) format, and then converted it into Core ML format. In order to execute the function expressions converted to Core ML format at high speed, an iPhone equipped with a high-performance neural engine, specifically the iPhone 11, iPhone SE2, or later (Apple Inc., Los Altos, CA, USA) is required. In this study, the iPhone 12 with an Apple A14 Bionic chip and the iPhone SE2 with an A13 Bionic chip were used. The A14 Bionic chip features a 64-bit 6-core central processing unit (CPU), 4-core graphics processing unit (GPU), and 16-core neural engine which can perform 11 trillion operations per second, and the A13 Bionic chip features a 6-core CPU, 4-core GPU, and 8-core neural engine.

### 2.3. Deep Learning for 3D Human Motion Estimation

As shown in Figure 2, three consecutive captured images were simultaneously used for input training data in order to learn the recognition of motion images, not simple static images. The loss function was set to the sum of L2 loss for the four outputs of the estimated 3D coordinates, as the learning condition of modified ResNet34, and Adam, a method for stochastic optimization, was set at a learning rate of 0.00001–0.000001. To estimate the relative 3D coordinate of human 3D motion, we adapted the 3D heatmap method, which consists of 28 × 28 × 28 blocks for the 24 key points with three RGB color (Figure 3). The most probable (hottest) block in each key point was estimated by the 3D heatmap and matched with the already extracted key points of the previous two frames. Additionally, in the hottest block of the 3D heatmaps, the final X-, Y-, and Z-coordinates in each key point were determined after making the fine corrections (deviating from the center of the block) calculated by the hyperbolic tangent (tanh) function as a nonlinear activation function. Once the output features (analysis pipeline) were extracted using the final convolution layers, they were mapped using a subset of fully connected layers to the final outputs, and the activation function applied to the multiclass task normalized real output values to target class probabilities, where each value ranges between 0 and 1, and the sum of all values was 1.

**Figure 2 sensors-22-05282-f002:**
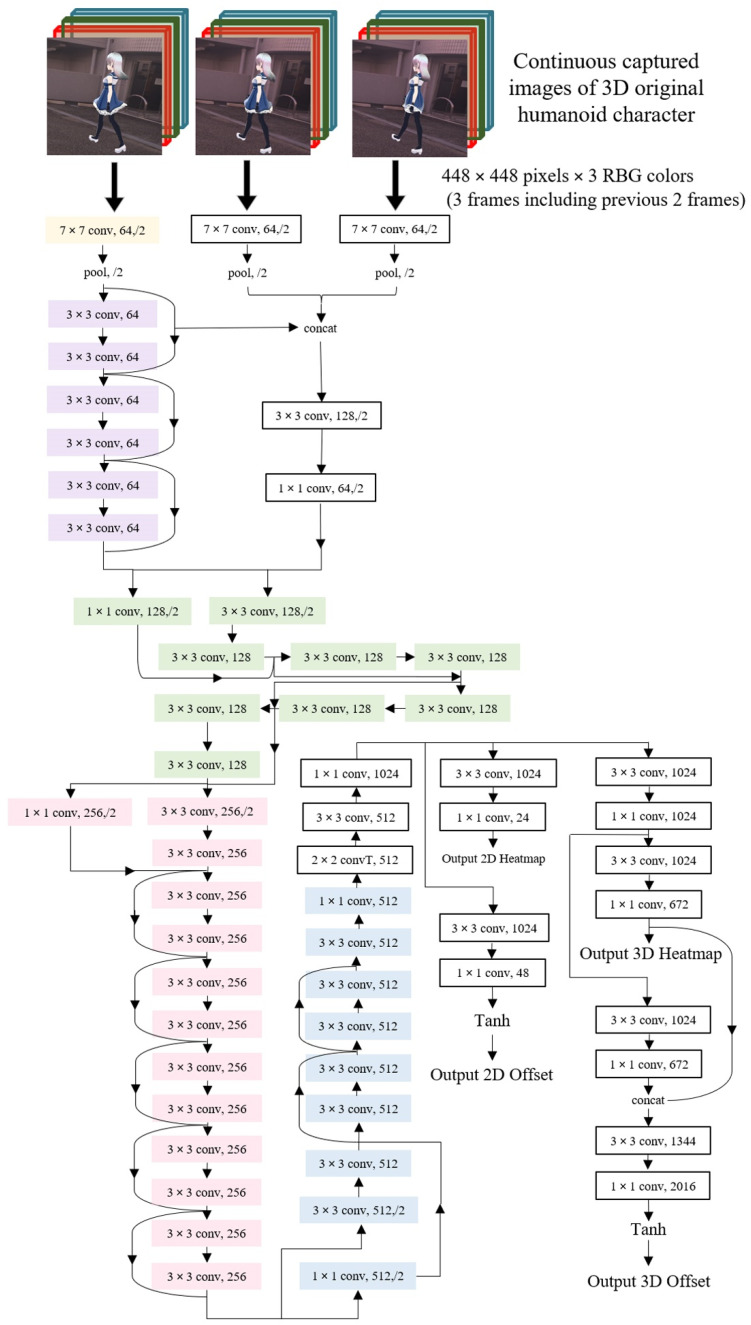
Overview of the training process and backbone network for the pose estimation framework. The colored cells are the basic cells designed using ResNet34, and the white cells are the added cells.

## 3. Results

A new and free iOS application named Three-Dimensional Pose Tracker (TDPT) has been released on the Apple store (https://apps.apple.com/app/tdpt/id1561086509?mt=8, accessed on 29 May 2022) for general users. TDPT can recognize a person in a 2D videoframe taken with a monocular iPhone camera and track the motion of the entire body in real time at high speeds (https://digital-standard.com/tdptios_en/, accessed on 29 May 2022). This application does not require access to other servers or libraries and preprocessing to obtain the relevant information required for the input query; users must only put the entire body from the head to the toe in the videoframe of the iPhone camera.

For the 3D motion analysis research, we modified the TDPT application to obtain the relative 3D coordinates of the 24 key points in the human full-body. This modified application named “TDPT for Gait Test (TDPT-GT)” has not been released. Start the TDPT-GT application, set the iPhone so that the full-body of the subject includes within the videoframe, and just press the button of start/stop to automatically save the estimated 3D relative coordinates between start and stop at 30 fps in the iPhone as a csv file (a sample output file in the Appendix A).

### 3.1. How to Use the TDPT for Gait Test Application

#### 3.1.1. Shooting Direction

Although the TDPT-GT application can easily estimate the relative 3D coordinates of the 24 key points in the human body during gait on the iPhone alone, video recording from the frontal side alone could not correctly capture the joint angles of the neck, lumbar, hip, knee, and ankle, while video recording from the lateral side alone misidentified the right and left sides of the limbs. Therefore, video recording from various directions of the human body during motion was needed to estimate the 3D coordinates correctly. Additionally, to confirm the deviation of the left and right joint angles because of the orbital walking movements, the subjects walked around a circle clockwise and counterclockwise. Finally, we took a video recording of walking around a circle with a diameter of 1 m for two laps, clockwise and counterclockwise (Figure 4 and Appendix A). 

#### 3.1.2. Precautions for Use

Although many acquired 3D coordinates of the 24 key points from the head to the toes using the TDPT-GT application were sufficiently robust enough to be used as a quantitative assessment, reliable coordinates of some key points could not be acquired. The acquired 3D relative coordinates were unreliable in the following situations: some parts of the body were out of the videoframe or reflected by the polished floor or mirror; the subject was far from the iPhone and appeared small in the videoframe; other people entered the videoframe; the clothes were loose or oversized, making the body line undetectable; and the lighting was dark.

### 3.2. Data Processing for Gait Analysis

For each key point, the 3D coordinates observed were raw, and those smoothed by applying a low-pass filter were automatically saved at 30 fps in the iPhone in a csv file format (Appendix A). The joint angles were calculated from the smoothed 3D coordinates. The midpoint of both shoulders was defined as the neck, and the midpoint of both ears was defined as the center of the head. The 3D angle of the hip joint was defined as the angle between the vector connecting the hip joint and navel (center of the body) and the vector connecting the hip and joints. The knee angle was defined as the angle between the vector connecting the knee and hip joints and the vector connecting the knee and ankle joints. The ankle angle was defined as the angle between the vector connecting the ankle and knee joints and the vector connecting the ankle joint and toe. 

Figure 5 shows the chronological changes in the 3D angles of the bilateral hip, knee, and ankle joints calculated from the 3D relative coordinates estimated by the TDPT-GT application for the normal gait shown in Figure 4 (Appendix A). 

The 3D angles of the right and left knee joints appeared to be estimated relatively accurately, with the left and right alternating in the range of 0 to 40 degrees during gait, whereas the angles of the hip and ankle joints were relatively inaccurate.

### 3.3. Verification of 3D Relative Coordinates on TDPT for Gait Test Application

To verify the reliability of the 3D relative coordinates measured by the TDPT-GT, the 3D coordinates were simultaneously measured by the Vicon Motion System (Oxford, UK), as a preliminary study. Because the local coordinate system in the TDPT-GT application and the global coordinate system in the Vicon have quite different standards for spatial coordinates, rotational movements were employed in the preliminary validation study rather than walking, which is more sensitive to spatial coordinates. In the rotational movements, the subject stayed in one place and turned slowly with the upper limbs extended (Appendix A and Figure 6). The plots shown on the right side of Appendix A (Figure 6) represent the time-axis-adjusted 3D coordinate transitions of the key points of the whole body during rotational movements with the TDPT-GT application and Vicon. The mean values of the 3D coordinates for the right and left shoulders, elbows, wrists, hip joints, knees, ankles, and toes measured by the TDPT-GT application and Vicon during rotational movements and Pearson’s correlation coefficient (*r*) for each coordinate are shown in Table 1.

## 4. Discussion

We developed a novel iOS application for a markerless motion capture system that estimates 3D human body poses from 2D video images of a person using an iPhone camera. Human motion capture is defined as a process of digitally recording the movements of a person by tracking pixels. Recently, motion tracking technology using deep learning has been subject to rapid progress. OpenPose, an open-source system for multi-person 2D pose estimation using Part Affinity Fields, has been applied to estimate the 3D coordinates [22]. Although the method of lifting 2D joint position coordinates to 3D spatial coordinates using the convolutional neural network is a great idea, this method requires specialized knowledge and is suitable for researchers. On the other hand, our newly developed application dose not need to extract image data from the smartphone, and users do not need to provide new training data; coordinate estimation is completed inside the smartphone. ResNet has been known as an innovative deep convolutional neural network applied in image recognition tasks. ResNet achieved top results for object detection and object detection with localization tasks in the competition of ImageNet Large Scale Visual Recognition Challenge (ILSVRC) 2015 [25]. As the number of layers of the neural network increases, the accuracy levels may become saturated and slowly degrade after a point. ResNet overcame this limitation by using residual blocks and skipping connections to improve accuracy and reduce training time. Our method is novel in image recognition technology using the state-of-the-art ResNet model in that three consecutive 2D images were used for deep learning as movies rather than static images capturing a moment of motion. Furthermore, we could prepare sufficient training 3D datasets using the original humanoid CG characters with skeletal information created by ourselves to learn human motion. In addition, we used the novel technology of 3D heatmaps with regression models proposed by Sun et al. [26]. The validity and effectiveness of this method has been convincingly verified in various settings, particularly comprehensive ablation experiments with 3D pose estimates [26]. They employed a small set of 2D images tracked from a monocular video sequence to reconstruct 3D motion. The advantage of the 3D heatmap method is that it visualizes the region in which the network focuses as an attention map, which can clarify the reason for the network decision-making in a visual explanation. The visual explanations enable us to analyze and understand the internal states of convolutional neural networks, which is efficient for engineers and researchers.

The greatest advantage of our developed smartphone application is that any clinician or researcher can easily obtain the relative 3D coordinates of the human whole-body without any markers using only a smartphone. Optical 3D motion capture systems with multipoint cameras and markers for motion capture are prohibitively expensive and may be unsuitable for universal clinical use. In contrast, our developed iOS application can be easily applied to human motion analysis by anyone without considerable amounts of time and money and does not require a laboratory setting. Furthermore, in our application, large amounts of human motion data can be processed within a few milliseconds. This enables the application to perform in real time to support capturing the 3D pose, analyzing gait, and encouraging physical exercise. Consequently, these measurements can be used in multicenter collaborative studies to assess the severity and changes in gait disturbances using a monocular smartphone camera.

This study has some limitations that warrant discussion. First, the reliability and validity of the 3D relative coordinates estimated by the TDPT-GT application have not yet been fully verified in this study. The 3D relative coordinates on the TDPT-GT are completely different from the 3D spatial coordinates on the Vicon in the global coordinate system. The 3D relative coordinates on the TDPT-GT were normalized as 1 for the length of the upper body from the navel to the center of the head, and 1 for the length of the lower body from the navel to the ankle, regardless of the subject’s height. There are still challenges in verifying the accuracy of the TDPT-GT application, such as the impossibility of aligning the time by heel contact and different scales for the upper and lower body, but the validation study is ongoing in our study group. Second, this system could track the body pose of one person only, and the iPhone camera must always show the entire body. Therefore, it may not be usable in a cluttered environment, such as a group health check or an outpatient examination room. Finally, the 3D relative coordinates based on the center of the body without the ground information in the local coordinate system were rarely used in previous gait studies. Therefore, the interpretation based on the obtained 3D spatial coordinates of the entire human body pose is different from that in previous studies. For instance, separating the stance phase from the swing phase or measuring the distance from the floor surface as in the conventional gait analysis is impossible. Further work is needed to determine the advantages and disadvantages of this new gait research, which performs analyses without specifying the floor surface.

## 5. Conclusions

We have developed a novel application that could obtain the 3D relative coordinates of the entire human body based on simple 2D video images of a person using a monocular iPhone camera. Our developed markerless monocular motion capture application could reconstruct the full-body human motion efficiently in real time. Using this application, anyone can easily perform 3D gait assessments quantitatively, which previously required a large-scale 3D motion analysis system, and we believe that it will be adopted in evaluating the clinical outcomes of surgery or rehabilitation for various movement disorders in clinical studies. Quantitative measurement utilizing a smartphone makes a wider range of people in well-being, comfort, and health monitoring.

## Figures and Tables

**Figure 1 sensors-22-05282-f001:**
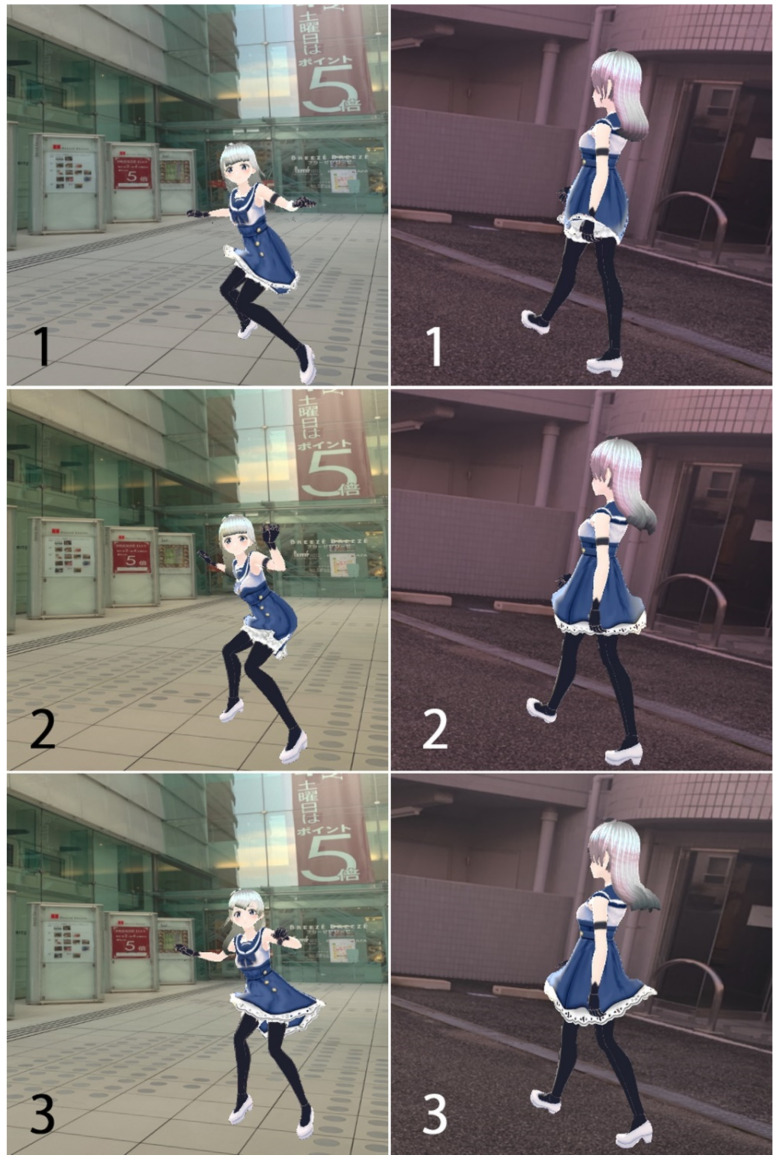
Sample of the input training data for deep learning. The three consecutive images on the left (black 1–3) are dancing movements, and those on the right (white 1–3) are walking movements.

**Figure 3 sensors-22-05282-f003:**
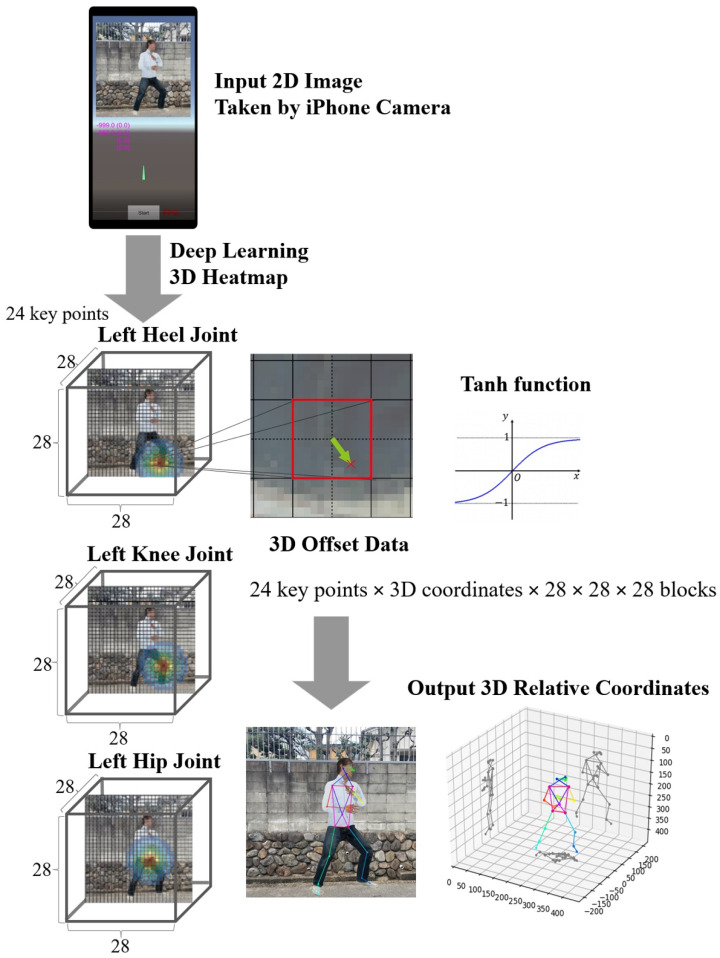
Output 3D relative coordinates of 24 key points on 3D heatmaps.

**Figure 4 sensors-22-05282-f004:**
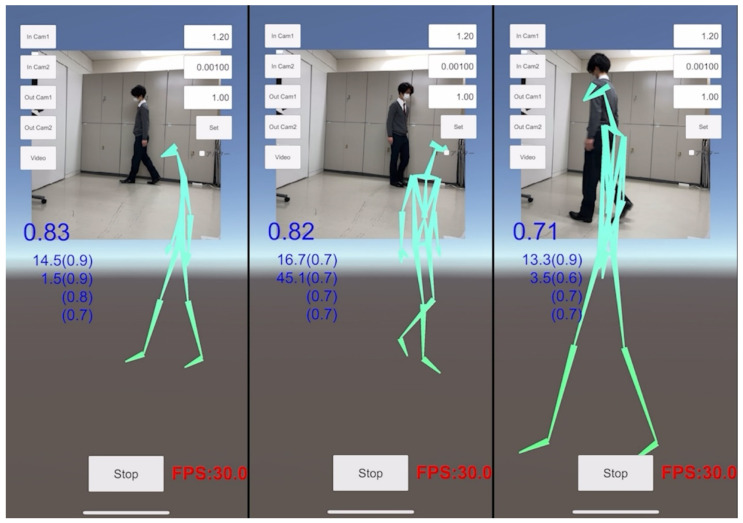
Snapshots of the TDPT for Gait Test (TDPT-GT) application and calculated 3D joint angles and artificial intelligence (AI) scores, also called the confidence scores in a healthy young volunteer. The iPhone was fixed as horizontal as possible to the floor. The entire body of the subject from the head to the toes must always be in the videoframe. The blue-colored number at the left of the videoframe shows the reliability of the position information of the entire body as an AI score, and the small numbers below show the 3D angles (AI scores) of the left and right knee joints, and (AI scores) of the left and right ankle joints.

**Figure 5 sensors-22-05282-f005:**
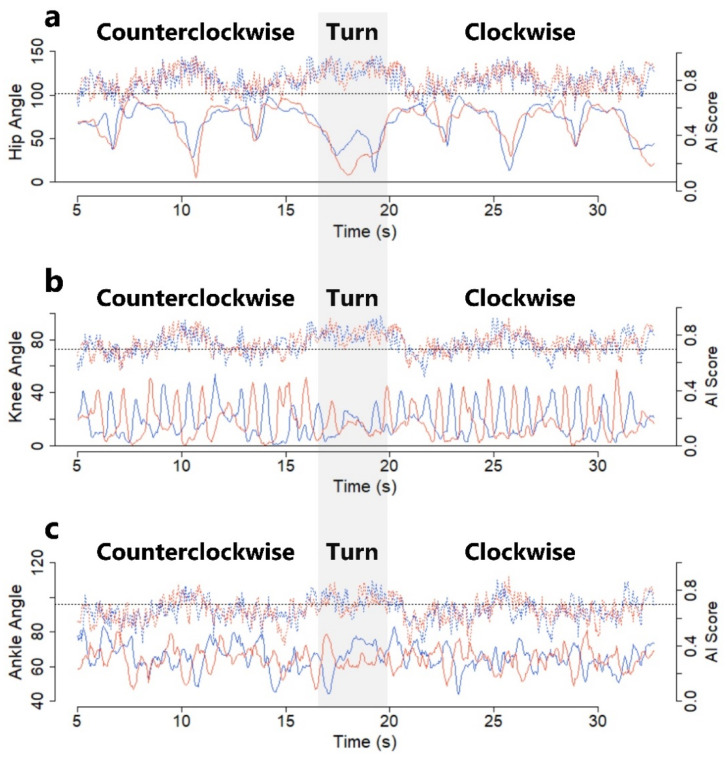
Three-dimensional angles (solid lines) calculated by the relative 3D coordinates estimated by the TDPT for Gait Test (TDTP-GT) application and artificial intelligence (AI) scores (dotted lines). When each AI score is 0.7 (black dotted line) or higher, the 3D angles at the right (blue) and left (red) hip joints (**a**), knee joints (**b**), and ankle joints (**c**) are considered to be relatively reliable.

**Figure 6 sensors-22-05282-f006:**
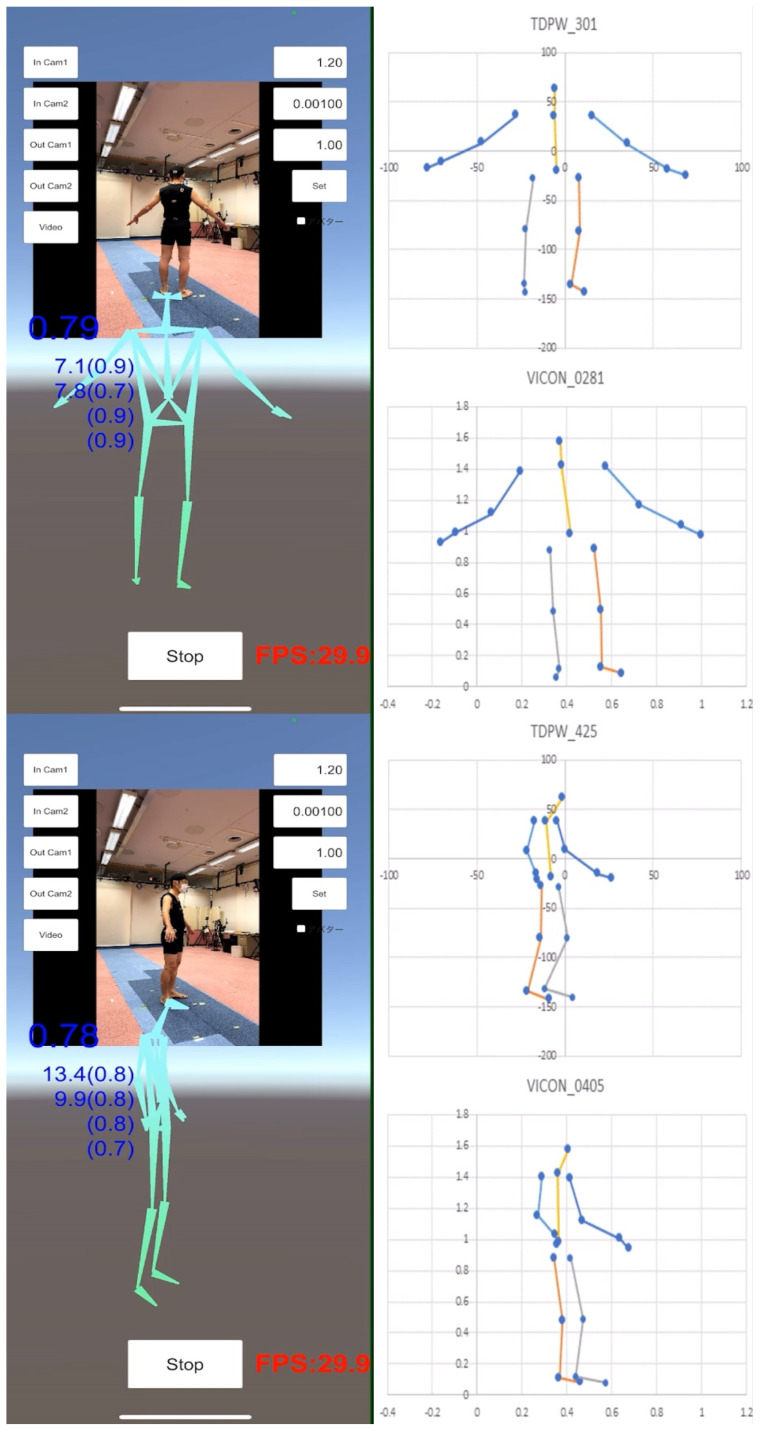
Simultaneous measurement by two methods of TDPT for Gait Test (TDPT-GT) application and Vicon Motion System. Blue dots indicate 3D coordinates; blue lines indicate both upper extremities, yellow indicates the trunk, gray indicates the left lower extremity, and orange indicates the right lower extremity.

**Table 1 sensors-22-05282-t001:** Relationship between 3D coordinates with TDPT for Gait Test (TDTP-GT) application and those with VICON Motion System.

	TDPT-GT	VICON (×10)	*r*
Right Shoulder	(0.3, 35.3, 1.6)	(4.2, 14.0, 0.7)	(0.87, 0.58, −0.84)
Left Shoulder	(−7.3, 35.0, −6.4)	(3.8, 14.0, 2.2)	(0.90, −0.49, −0.34)
Right Elbow	(4.3, 8.1, 4.4)	(4.6, 11.5, 0.2)	(0.92, −0.12, −0.94)
Left Elbow	(−11.0, 7.3, −9.9)	(3.8, 11.4, 2.8)	(0.89, −0.36, −0.67)
Right Wrist	(4.6, −15.3, 8.7)	(4.4, 10.1, −0.7)	(0.95, −0.23, −0.89)
Left Wrist	(−17.5, −14.4, −12.0)	(3.4, 10.1, 3.4)	(0.87, −0.40, −0.82)
Right Hip joint	(−1.1, −27.3, 2.9)	(4.6, 8.8, 1.2)	(0.85, −0.02, −0.85)
Left Hip joint	(−5.9, −27.5, −2.0)	(4.3, 8.8, 2.0)	(0.79, 0.52, −0.29)
Right Knee	(−2.7, −79.1, 1.9)	(4.7, 4.8, 1.4)	(0.93, −0.77, −0.60)
Left Knee	(−8.3, −79.0, −4.0)	(4.5, 4.9, 2.1)	(0.77, 0.49, −0.19)
Right Ankle	(−3.3, −132.2, 4.4)	(5.2, 1.2, 1.6)	(0.84, −0.75, −0.88)
Left Ankle	(−7.2, −131.6, −2.4)	(4.9, 1.2, 2.2)	(0.81, 0.14, −0.61)
Right Toe	(−6.0, −140.3, 7.4)	(4.8, 0.7, 1.3)	(0.92, −0.66, −0.77)
Left Toe	(−11.1, −140.4, −0.6)	(4.4, 0.7, 2.2)	(0.74, 0.57, −0.72)

The mean values of 3D coordinates (X, Y, Z) for TDPT-GT and VICON (×10). *r*: Pearson’s correlation coefficient for each coordinate (X, Y, Z).

## Data Availability

Data generated or analyzed during the study are available from the corresponding author upon reasonable request.

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
