# Peer review of "Development of Smartphone Application for Markerless Three-Dimensional Motion Capture Based on Deep Learning Model"

_sensors, 2022, doi:10.3390/s22145282_

Round 1

Reviewer 1 Report

This paper briefly describes the development of a deep learning model for full-body human motion tracking in real-time from markerless video-based images using a smartphone monocular camera. The generated image dataset was used to evaluate the performance of the proposed model. In addition, a mobile application for iPhone users was developed to show the practical application of the proposed model. Overall, the study is quite interesting and is categorized as a brief report or communication type of research article. However, several modifications should be addressed, such as:

-The title could be simplified, such as "Development of mobile application based on deep learning model..."

-In the abstract, please reveal the results of the study (measured results) as stated in the beginning of the abstract, that this research was "quantitatively assessed."

- Even though this is a communication type of paper, please provide the necessity as well as the importance of this research problem in general and specific in the first and second paragraphs of the introduction section.

-Previous studies related to gait analysis should also be discussed in the introduction section.

-The summary contribution of the study should be presented at the end of the introduction section.

- It is unclear whether the analysis was performed inside the iPhone or through the remote server. Please clarify this by providing the flow-chart of the system from end to end (from user/iphone application to analysis (gait detection)).

-The servers and smartphones specifications as well as the libraries used for the applications (in servers and mobile applications) should be revealed in as much detail as possible to increase the usability and reproducibility of the study.

-The provided link of application in the results section is different from Figure 3. Please clarify.

-The proposed model should be "quantitatively assessed," but the current form did not reveal any measured metrics used in the study. Please provide them!

Author Response

We thank the reviewer for taking the time to further review our manuscript and provide valuable comments and suggestions for its further improvement. According to the reviewer’s comments, we have changed almost all parts of our manuscript including title and all figures after careful reading of instructions.

Q1. The title could be simplified, such as "Development of mobile application based on deep learning model..."

A1. We thank the reviewer for sincere comments. According to the reviewer’s suggestion, we changed the title as " Development of smartphone application for markerless three-dimensional motion capture based on deep learning model"

Q2. In the abstract, please reveal the results of the study (measured results) as stated in the beginning of the abstract, that this research was "quantitatively assessed."

A2. According to the reviewer’s pointed out, the entire abstract has been revised in line with this major revision of the text, in compliance with the 200-word limit.

Q3. Even though this is a communication type of paper, please provide the necessity as well as the importance of this research problem in general and specific in the first and second paragraphs of the introduction section.

A3. According to the reviewer’s suggestion, we added the necessity and importance of this research problem in the introduction section.

Q4. Previous studies related to gait analysis should also be discussed in the introduction section.

A4. We transferred the description of the previous studies related to gait analysis from the discussion section to the introduction section.

Q5. The summary contribution of the study should be presented at the end of the introduction section.

A5. According to the reviewer’s instructive advice, we added the summary of contribution and originality of the study at the end of the introduction section.

Q6. It is unclear whether the analysis was performed inside the iPhone or through the remote server. Please clarify this by providing the flow-chart of the system from end to end (from user/iphone application to analysis (gait detection)).

A6. We are sorry that the detailed explanation about “TDPT for Gait Test (TDPT-GT)” application was not enough. This application can estimate the relative 3D coordinates of the 24 key points on a stand-alone basis with only one iPhone, and does not require access to other servers or libraries. Start the application, set the iPhone so that the full-body of the subject includes within the videoframe of the application, and only press the button of start / stop to save the quantified 3D relative coordinates in the iPhone as a csv file automatically. We added the detailed explanation in the Materials and Methods and Results sections.

Q7. The servers and smartphones specifications as well as the libraries used for the applications (in servers and mobile applications) should be revealed in as much detail as possible to increase the usability and reproducibility of the study.

A7. As mentioned above, this application does not use servers and libraries. We added the detail information about the smartphones’ specifications in the Materials and Methods and Results sections, and the originality and strength of our application in the Introduction and Discussion sections.

Q8. The provided link of application in the results section is different from Figure 3. Please clarify.

A8. The provided link in the first paragraph of the results section is for Three-Dimensional Pose Tracker (TDPT) application for general users. The modified application named “TDPT for Gait Test (TDPT-GT)” has been used only for research purposes and has not been released, because we still have plans to make improvements, and the final form of the application release has not yet been finalized. However, for the readers, we added the source code of TDTP application published on GitHub (https://github.com/digital-standard/ThreeDPoseUnityBarracuda, https://github.com/yukihiko/ThreeDPoseUnitySample).

Q9. The proposed model should be "quantitatively assessed," but the current form did not reveal any measured metrics used in the study. Please provide them!

A9. According to the reviewer’s suggestion, we attached text for the input 3D training dataset sample of Figure 1 and csv file for the output 3D relative coordinates estimated by GT-TDTP application as Supplementary Materials.

Reviewer 2 Report

General Comments:

The methodology is sound and the result is prepared neatly. The construct of the paper overall is quite good. It seems that a significant amount of time and effort has been put into preparing this paper. However, there are some concerns regarding the content to which this manuscript is prepared.

  • Some figures appear to be not in the format of the sensors journal.
  • Details of the methodology and results

The authors developed an iOS app that can estimate the full-body segment orientation without using markers. Machine learning is implemented to heat maps and camera images of a subject to estimate the 3D body motion within a given closed volume of space. My main concern with this paper is the lack of comparison to other conventional means of motion capture. This paper goes into great detail about the methodology of the system but little in terms of quantifiable results. Statistically meaningful comparisons must be made with other conventional motion tracking systems to prove the validity to which the proposed system intends to replace. If this is not possible, comparing measurement conditions (image resolution, distance, motion speed, etc) within the same system is also possible.

Specific comments are below:

Line 40        

A more thorough introduction required

Line 62

“Therefore, this study was designed to develop a smartphone application that could replace 3D motion capture systems.”

However, no comparisons are made in the manuscript.

Line 67

“used the AI bionic chip neural engine”

Unfamiliar term. Please define what exactly this is and how relevant it is to this method. Include manufacturer and model number.

Eg. Faster processing power, specifically required for a certain development environment, etc.

Line 77

Fonts too small in Fig.1

Line 139

Graph and fonts too small in Fig.3

Line 162

To assess the applicability to gait analysis, various spatial-temporal gait parameters must be included. The graphs in fig.3 are not comprehensive. In addition, no information about the subjects is given.

Line 173

The discussion should include more quantitative result comparisons rather than qualitative ones.

Author Response

We thank the reviewer for taking the time to further review our manuscript and provide valuable comments and suggestions for its further improvement. According to the reviewer’s comments, we have changed almost all parts of our manuscript including title and all figures after careful reading of instructions.

Response:

We totally agree with the reviewer’s comments. The comparison with the gold standard method, optical motion capture system should be required to verify the certainty of 3D coordinate estimation by our developed application, but we have just started the verification study and have not yet fully analyzed. However, the greatest challenge is to convert the 3D spatial coordinates in the global coordinate system measured by the Vicon system to those in the local coordinate system, as measured by the TDPT-GT. Furthermore, TDPT-GT transforms the upper and lower body at different scales in 3D relative coordinates with respect to the body center (navel). Therefore, in this manuscript, we focused on the development of the TDPT-GT application that could obtain the 3D relative coordinates of the entire human body based on simple 2D video images of a person using a monocular iPhone camera.

However, we added one movie (Video S2, Fig. 6) and explanations in the results section, as a preliminary validation study, according to the reviewer’s suggestion.

Line 40       

A more thorough introduction required

Response:

We added the previous studies related to gait analysis, the necessity and importance of this research problem, and the summary of contribution of the study in the introduction section.

Line 62

“Therefore, this study was designed to develop a smartphone application that could replace 3D motion capture systems.”

However, no comparisons are made in the manuscript.

Response:

As mentioned above, we added the results of our preliminary validation study. As the reviewer point out, however, this wording was not appropriate. We changed as follows; “Therefore, this study was designed to develop a totally novel smartphone application that can directly estimate 3D coordinates of human motion from 2D images taken by the monocular camera of a smartphone using a novel deep learning model.”

Line 67

“used the AI bionic chip neural engine”

Unfamiliar term. Please define what exactly this is and how relevant it is to this method. Include manufacturer and model number.

Eg. Faster processing power, specifically required for a certain development environment, etc.

Response:

According to the reviewer’s advice, we changed and added the detail information in the Chapter 2.2. Execution environment in lines 168-174.

In order to execute the function expression converted to Core ML format at high speed, an iPhone equipped with a high-performance neural engine, specifically iPhone 11, iPhone SE2 or later (Apple Inc.) is required. In this study, the iPhone 12 with an Apple A14 Bionic chip and the iPhone SE2 with an A13 Bionic chip were used. The A14 Bionic chip features a 64-bit 6-core central processing unit (CPU), 4-core graphics processing unit (GPU), and 16-core neural engine which can perform 11 trillion operations per second, and the A13 Bionic chip features a 6-core CPU, 4-core GPU, and 8-core neural engine.

Line 77

Fonts too small in Fig.1

Response:

According to the reviewer’s suggestion, we changed Fig.1 (new Fig.2) to make the fonts as large as possible.

Line 139

Graph and fonts too small in Fig.3

Response:

According to the reviewer’s suggestion, we changed Fig.3 (new Fig.5) to make the graph and fonts as large as possible.

Line 162

To assess the applicability to gait analysis, various spatial-temporal gait parameters must be included. The graphs in fig.3 are not comprehensive. In addition, no information about the subjects is given.

Response:

We thank the reviewer for sincere comments. According to the reviewer’s suggestion, we changed Fig.3 (new Fig.4 and Fig.5). In new Fig.5, we show the 3D angles of bilateral hip joints, knee joints, and ankle joints during circle walking as shown in new Fig.4.

The man in the new Fig.3 is the first author (Y.A.), and the man in the new Fig.4 (and Fig.5) is the co-author (K.M.), and the man in the new Fig.6 is the co-author (Y.K.).

A study of extracting various kinematic information from the 3D relative coordinates obtained by this application is planned for the next report and beyond.

Line 173

The discussion should include more quantitative result comparisons rather than qualitative ones.

Response:

According to the reviewer’s suggestion, we entirely changed the Discussion section.

Reviewer 3 Report

Authors introduced the development of a novel phone camera based markerless motion capture system for gait analysis. However, there are several key concerns should be addressed, which are listed in below.

1. Line 27, what is the 'teacher data'?

2. Introduction, the background information in the current form is not enough to justfy this study, and suggest separating into different paragraphs to highlight the key points from literature.

3. Line 70-73, please use short sentence.

4. Line 114-116, this statement is more of methodology, please move into method section.

5. Results, this section for now is quite simple, and no results concerning the joint anlges from phone camera was reported and analysed, and compared against the gold-standard motion capture system.

6. The Discussion section should be rewritten accordingly considering the points raised above. 

Author Response

We thank the reviewer for taking the time to further review our manuscript and provide valuable comments and suggestions for its further improvement. According to the reviewer’s comments, we have changed almost all parts of our manuscript including title and all figures after careful reading of instructions.

Response:

Thank you for instructive comments and suggestions. We changed all parts of our manuscript including the title and references.

Comments and Suggestions for Authors

Authors introduced the development of a novel phone camera based markerless motion capture system for gait analysis. However, there are several key concerns should be addressed, which are listed in below.

Q1. Line 27, what is the 'teacher data'?

A1. It’s a typo, and corrected as 'training data'.

Q2. Introduction, the background information in the current form is not enough to justify this study, and suggest separating into different paragraphs to highlight the key points from literature.

A2. We added the previous studies related to gait analysis, the necessity and importance of this research problem, and the summary of contribution of the study in the Introduction section.

Q3. Line 70-73, please use short sentence.

The convolutional neural network was the most established algorithm among various machine learning models for processing data that have a grid pattern; however, plain models of the convolutional neural network have a degradation problem, that is, poor training accuracy as the depth of the network increases.

A3. According to the reviewer’s suggestion, at "however", we divided it into two sentences.

Q4. Line 114-116, this statement is more of methodology, please move into method section.

A4. According to the reviewer’s suggestion, the first sentence was moved into the Materials and Methods section. In addition, we changed and added the detail information in the Chapter 2.2. Execution environment in lines 168-174.

In order to execute the function expression converted to Core ML format at high speed, an iPhone equipped with a high-performance neural engine, specifically iPhone 11, iPhone SE2 or later (Apple Inc.) is required. In this study, the iPhone 12 with an Apple A14 Bionic chip and the iPhone SE2 with an A13 Bionic chip were used. The A14 Bionic chip features a 64-bit 6-core central processing unit (CPU), 4-core graphics processing unit (GPU), and 16-core neural engine which can perform 11 trillion operations per second, and the A13 Bionic chip features a 6-core CPU, 4-core GPU, and 8-core neural engine.

Q5. Results, this section for now is quite simple, and no results concerning the joint angles from phone camera was reported and analyzed, and compared against the gold-standard motion capture system.

A5. As the reviewer’s suggestion, we focused on the development of the TDPT-GT application that could obtain the 3D relative coordinates of the entire human body based on simple 2D video images of a person using a monocular iPhone camera in this manuscript. That’s the main result.

According to the reviewer’s point out, we added the 3D angles of bilateral hip joints, knee joints, and ankle joints during circle walking in new Fig.5, and the one movie (Video S2, Fig. 6) and explanations in the Results section, as a preliminary validation study.

Q6. The Discussion section should be rewritten accordingly considering the points raised above.

A6. According to the reviewer’s suggestion, we entirely changed the Discussion section.

Round 2

Reviewer 1 Report

There are no any concerns with the paper as they addressed all of the earlier comments.

Author Response

Thank you for your adequate peer review.

Reviewer 2 Report

Though the authors made significant changes to the manuscript, I still believe that a comparison with other motion analysis systems is necessary to point of the merits of the proposed system.

Author Response

In our previous reply to the comments from the reviewer, we had precisely described the preliminary validation results in the new chapter 3.3, and added new Video S2 and Figure 6. However, as described the first limitation in the discussion section, The 3D relative coordinates on the TDPT-GT are completely different from the 3D spatial coordinates on the Vicon in the global coordinate system. There are still challenges in verifying the accuracy of the TDPT-GT application such as impossible to align the time by heel contact and different scales for the upper and lower body, but the validation study is ongoing in our study group.

The comparison method the Reviewer proposed is not a verification of the reliability of the 3D conditions measured by TDPT-GT application, but simply which condition can capture the coordinates most accurately, which we have already tried in various situation and described in the chapter 3.1.2. Precautions for use.

The previous resubmission was required within 10 days, this time within 5 days, and no further additional experiments can be performed. We can’t answer any further about this.

Reviewer 3 Report

Authors have made subtantial revisions and the quality of this manuscript is improved significantly.

Author Response

(The authors gave the same response as above.)
